# Imidazo-Pyrazole-Loaded Palmitic Acid and Polystyrene-Based Nanoparticles: Synthesis, Characterization and Antiproliferative Activity on Chemo-Resistant Human Neuroblastoma Cells

**DOI:** 10.3390/ijms241915027

**Published:** 2023-10-09

**Authors:** Giulia Elda Valenti, Barbara Marengo, Marco Milanese, Guendalina Zuccari, Chiara Brullo, Cinzia Domenicotti, Silvana Alfei

**Affiliations:** 1Department of Experimental Medicine (DIMES), University of Genova, Via Alberti L.B., 16132 Genoa, Italy; giuliaelda.valenti@edu.unige.it (G.E.V.); barbara.marengo@unige.it (B.M.); 2Department of Pharmacy, Section of Chemistry and Pharmaceutical and Food Technologies, University of Genoa, Viale Cembrano, 4, 16148 Genoa, Italy; marco.milanese@unige.it; 3Department of Pharmacy (DIFAR), Section of Medicinal Chemistry and Cosmetic Product, University of Genoa, Viale Benedetto XV, 3, 16132 Genoa, Italy; guendalina.zuccari@unige.it (G.Z.); chiara.brullo@unige.it (C.B.)

**Keywords:** human neuroblastoma (NB), etoposide (ETO)-sensitive cells, ETO-resistant cells, cationic polystyrene-based copolymer (P5), imidazo-pyrazoles (IMPs), palmitic acid (PA), reactive oxygen species (ROS)

## Abstract

Neuroblastoma (NB) is a childhood cancer, commonly treated with drugs, such as etoposide (ETO), whose efficacy is limited by the onset of resistance. Here, aiming at identifying new treatments for chemo-resistant NB, the effects of two synthesized imidazo-pyrazoles (IMPs) (**4G** and **4I**) were investigated on ETO-sensitive (HTLA-230) and ETO-resistant (HTLA-ER) NB cells, detecting **4I** as the more promising compound, that demonstrated IC_50_ values lower than those of ETO on HTLA ER. Therefore, to further improve the activity of **4I**, we developed **4I**-loaded palmitic acid (PA) and polystyrene-based (P5) cationic nanoparticles (P5PA-4I NPs) with high drug loading (21%) and encapsulation efficiency (97%), by a single oil-in-water emulsification technique. Biocompatible PA was adopted as an emulsion stabilizer, while synthesized P5 acted as an encapsulating agent, solubilizer and hydrophilic–lipophilic balance (HLB) improver. Optic microscopy and cytofluorimetric analyses were performed to investigate the micromorphology, size and complexity distributions of P5PA-4I NPs, which were also structurally characterized by chemometric-assisted Fourier transform infrared spectroscopy (FTIR). Potentiometric titrations allowed us to estimate the milliequivalents of PA and basic nitrogen atoms present in NPs. P5PA-4I NPs afforded dispersions in water with excellent buffer capacity, essential to escape lysosomal degradation and promote long residence time inside cells. They were chemically stable in an aqueous medium for at least 40 days, while in dynamic light scattering (DLS) analyses, P5PA-4I showed a mean hydrodynamic diameter of 541 nm, small polydispersity (0.194), and low positive zeta potentials (+8.39 mV), assuring low haemolytic toxicity. Biological experiments on NB cells, demonstrated that P5PA-4I NPs induced ROS-dependent cytotoxic effects significantly higher than those of pristine **4I**, showing a major efficacy compared to ETO in reducing cell viability in HTLA-ER cells. Collectively, this **4I**-based nano-formulation could represent a new promising macromolecular platform to develop a new delivery system able to increase the cytotoxicity of the anticancer drugs.

## 1. Introduction

Neuroblastoma (NB) is a cancer of immature nerve cells representing the most common extracranial solid tumour in childhood [1]. In the USA, about 700–800 new cases are detected per year, with about 1 case per 7000 live births [2,3,4]. Particularly, the occurrence is 10.2 cases per 1 million per year in children under 15 years old. The overall incidence in children up to 15 years of age is approximately 15,000 new cases per year in the world, of which approximately 130 are in Italy [2].

Generally, NB is rare in people over the age of 10 years and rarely it is detected by ultrasound even before birth [3]. About 37% of patients are diagnosed as infants, and 90% are younger than 5 years at diagnosis, with a median age at diagnosis of 17 months [4,5], thus establishing that NB is a disease of infancy, with the highest rate of diagnosis in the first month of life [6]. Treatments for NB patients currently include surgery, to remove non-metastatic cancer, chemotherapy, radiation therapy and treatment using iodine 131-MIBG to kill the cancer cells of primary localized NB, while high-dose chemotherapy and radiation therapy with stem cell rescue are used to kill any post-treatment residual cancer cells that may cause cancer relapse [7]. Additionally, a targeted therapy, using drugs or other substances to identify and attack specifically cancer cells, can be adopted in case of relapse or when NB is not responsive to other treatments [7,8]. Furthermore, new types of treatment, including immunotherapy, are being tested in clinical trials, and for some patients taking part in a clinical trial could be the best choice of treatment [7,8]. Table 1 collects some therapeutic approaches currently used to treat NB.

Unfortunately, NB treatments cause severe side effects [12] and following long-term treatments, the onset of therapy resistance is often inevitable and leads to treatment failure and cancer relapse [13]. Moreover, despite advances in the molecular aspect of paediatric cancers, the treatment of approximately 50% of children with high-risk neuroblastoma (HR-NB) lack efficacy [14], and current therapies are not operative in the long-term treatment of almost 80% of patients with this clinically aggressive form [15]. Also, when therapies are successful, survivors remain at risk for a wide variety of potential treatment-related complications, or “late effects”, which may lead to excess morbidity and premature mortality [8,16]. As an example, ETO, a DNA damage-inducing agent widely used to treat NB, can have some long-term side effects, including increased risk of developing secondary cancers, such as leukaemia or myelodysplastic syndrome. In this scenario, alternative therapeutic strategies are urgently necessary, both to counteract the development of chemoresistance and to reduce the side and late effects of current available treatments [17]. In this context, the Neuroblastoma New Drug Development Strategy (NDDS) is an initiative commenced in 2012 to accelerate the development of new drugs for NB [18]. Since drugs at nanosized dimensions reduce toxicity, prolong half-life, and increase the permeability of not nano-formulated compounds, our aim was to develop a novel possible therapeutic agent for NB treatment using a nanotechnological approach to formulate imidazo-pyrazole derivatives, which have previously demonstrated an interesting antiproliferative activity on several tumour cell lines [19]. To this end, we first screened the synthesized imidazo-pyrazole derivatives (IMPs) **4G** and **4I** (Figure 1) [19], on ETO-sensitive (HTLA-230) and ETO-resistant (HTLA-ER) NB cells.

Secondly, by a single oil-in-water nano-emulsion method, we employed the more active derivative (**4I**) to prepare **4I**-loaded NPs using palmitic acid (PA) as an emulsion stabilizer and the synthesized cationic polystyrene-based copolymer (P5) as both an encapsulating and solubilizing macromolecule (Figure 2).

P5PA-4I NPs were achieved, which were completely characterized by several analytical techniques and biologically tested on HTLA-230 and HTLA-ER cells. In parallel, NPs not containing **4I** (P5PA NPs) were prepared, characterized and tested on NB cells, in the same conditions used for P5PA-4I NPs.

## 2. Results and Discussion

### 2.1. The Reasons of Our Choice

We selected PA (Figure 2a) as a stabilizer, because it is a biocompatible saturated fatty acid (FA), with a C16 chain found in animal and vegetable oils, which was reported for having intrinsic anti-tumour activity in a variety of tumour types, including colorectal, liver, prostate and breast cancer [20,21,22,23,24,25], as many other exogenous FAs, which were effective in overcoming drug resistance and metastasis in tumour cells [26,27].

In this regard, Urso and Zhou have reported that PA is lipotoxic also to Neuro-2a (N2a) neuroblastoma cells by causing cell cycle defects [28]. Unfortunately, the poor water solubility and cellular impermeability of PA strongly limit its clinical application [29]. Very recently, such restrictions were addressed by the group of He, encapsulating PA into a poly (D, L-lactic co-glycolic acid) (PLGA) nanoparticle (NP) platform, alone and in combination with doxorubicin (DOX), achieving PLGA-PA-DOX NPs and PLGA-PA NPs, which significantly reduced the viability and migratory capacity of breast cancer cells in vitro [29]. Additionally, in in vivo studies in mice models of mammary tumours, the authors demonstrated that PLGA-PA-NPs were effective in reducing primary tumour growth and metastasis as NPs loaded with DOX, PA and DOX, or free DOX [29]. Moreover, Andrés González-González and colleagues demonstrated that the interaction of PA and SiNPs had a strong synergistic character in terms of emulsion stabilization, leading to an enhanced emulsion stability [30]. On these examples, in place of commercial PLGA utilized by He et al. [29] or SiNPs used by González-González et al. [30], we used the cationic P5 NPs (Figure 2b) [31] dispersed in water in association with PA dissolved in the oily phase with or without **4I**, to prepare both P5PA and **4I**-loaded P5PA oil-in-water (O/W) nano-emulsions stabilized by PA. Specifically, PA was able to interact with P5 dispersed in water by mean of hydrogen bond interactions [30]. Notably, P5 was selected also due to its intrinsic anti-cancer effects reported on HTLA-230, HTLA-ER [32], as well as on IMR-32 and SHSY 5Y NB cells [33].

### 2.2. Screening of the Cytotoxic Effects of ***4G*** and ***4I*** on NB Cells

The first aim of this study was to evaluate the cytotoxicity of **4G** and **4I** on HTLA-230, an ETO-sensitive NB cell line and HTLA-ER, the ETO-resistant counterpart [34]. As shown in Figure 3, **4G** exerted a dose- and time-dependent cytotoxic effect on both cell populations. In contrast, **4I** induced a more marked cytotoxic effect, especially in HTLA-ER, that reached a plateau level at 5 µM, thus resulting as more effective than **4G** (Figure 4).

Moreover, **4G** was capable of reducing cell viability by about 45-50% after 72 h, at 8-10 µM concentration in HTLA-230 cells and 20 µM in HTLA-ER. Differently, **4I** induced a similar cytotoxic effect in both cell populations at 5 µM, thus further supporting that **4G** was less effective than **4I** (Figure 4). By using GraphPad Prism 5.4.2 Software (GraphPad Software, Boston, MA, USA), and the mean data showed in Figure 3 and Figure 4, we obtained the values of IC_50_ for both **4G** and **4I.** As observed in Table 2, only the IC_50_ values at 48 and 72 h of exposure have been reported since at 24 h the cell viability was >50% at all concentrations tested.

Data reported in Table 2 confirmed that the cytotoxic effects of **4G** and **4I** were marked after 72 h, with **4I** more active than **4G**. According to data previously reported [19], both compounds would exert cytotoxic effects on tumour cells affecting the cell-cycle phases with the appearance of polyploid cells, altering microtubules and targeting the tubulin system. The higher activity of **4I** with respect to **4G** observed here could consequently depend on the different ways the two compounds interact with the tubulin system [19]. Particularly, it seems that although the shorter chain of **4G** allows the molecule to fit into a small cleft formed by tubulin β and stathmin3, thus reinforcing the binding between the two proteins, molecule **4I** is positioned in a way that might prevent the correct binding of tubulin tyrosine ligase to the tubulin α chain [19]. Given that, in HTLA-230, **4I** and **4G** had IC_50_ values of 10.78 and 12.31 µM, respectively, and that 10 µM IC_50_ has been retained by the National Cancer Institute (NCI), the threshold useful to qualify a new product as active in terms of in vitro antitumor activity, **4I** and **4G** can be considered as promising new agents to develop novel formulations to cure HR-NB. Although, in HTLA-ER, the IC_50_ values were 2-fold higher than the threshold dictated by NCI, **4I** was 1.8-fold more active than ETO. Based on this evidence, the following biological assays were carried out exclusively with **4I** with the aim to investigate the possibility of enhancing the 4I activity by a nano-formulation approach.

### 2.3. Dose Dependent Reactive Oxygen Species (ROS) Production in HTLA-230 and HTLA-ER Cells

ETO, as several chemotherapeutic drugs, exerts an anticancer effect mainly via ROS production and oxidative DNA damage [34,35,36]. In this context, it has been reported that glutathione-mediated antioxidant response has a pivotal role in determining the chemoresistance of HR-NB cells [34]. Therefore, to investigate if the cytotoxic effects of **4I** could depend on ROS production, the levels of H_2_O_2_, the most long-lived ROS, were analysed and the results have been reported in Figure 5.

DCFH-DA staining is a simple and cost-effective way to detect ROS in cells after chemical treatment or genetic modifications [37]. As shown in Figure 5, in HTLA-230, a significant increase in H_2_O_2_ production was observed at concentrations ≥1 µM, both after 24 h of exposure (statistical significance *** vs. Ctr) and 48 h (statistical significance **** vs. Ctr), while at concentration 0.5 µM (statistical significance ** vs. Ctr), after 72 h of exposure. Anyway, when HTLA-230 cells were exposed to **4I** for 24 h, the H_2_O_2_ production remained constant for concentrations ≥8 µM. On the contrary, in HTLA-ER cells, after 24 h of treatment, no significant variation of H_2_O_2_ production was observed in the range of concentrations tested. A significant production of H_2_O_2_ (statistical significance ** vs. Ctr) was observed in HTLA-ER cells treated with **4I** for 48 and 72 h at concentrations ≥1 µM. However, while H_2_O_2_ production increased further for concentrations ≥1 µM after 72 h of exposure, it remained constant after 48 h of exposure for concentrations ≥5 µM.

Moreover, in HTLA-230 cells treated with 10 µM **4I** for 72 h, H_2_O_2_ levels were 2-fold higher than in the control cells. In this contest, Nambiar et al. demonstrated that the antiproliferative mechanism of an analogue of noscapine in different tumour cell lines, depended on the alteration of the tubulin system that in turn depended on high ROS levels (2-3-fold higher than in the control) [38]. As mentioned above, according to Nambiar, Brullo et al. demonstrated that **4I** altered the microtubules [19] even if they did not investigate the effects of **4I** on ROS levels. Based on these findings, we can speculate that **4I**-treated HTLA-230 cells die in consequence of the microtubule alterations [19] that are probably triggered by the increased ROS production in agreement with the study of Nambiar et al. [38].

#### Correlation between ROS Production and Cytotoxic Effects of **4I**

To confirm the possible mechanism hypothesized in Section 2.2, we correlated data of cytotoxicity with those of ROS production by plotting cell viability (%) vs. DCFH positive cells (%). Figure 6a (HTLA-230) and Figure 6b (HTLA-ER) show the obtained dispersion graphs at 24, 48, and 72 h of exposure.

The values of R^2^ of the linear regressions of all dispersion graphs, obtained by the Ordinary Least Squares (OLS) method, offered information about the correlation existing between ROS levels and cell death. Specifically, a very good correlation was found in HTLA-230 cells at all times of exposure, confirming that **4I** acted mainly by enhancing ROS production, whose effects were cytotoxic for ETO sensitive cells displaying insufficient antioxidant defences. With regard to HTLA-ER cells, **4I**-induced cytotoxic effects were very low, especially at 24 h according to ROS decrease, probably due to the high amount of glutathione in these cells [34]. As evidenced by the very low value of R^2^ of the linear regression in Figure 6b (purple dispersion), the minimal cell death was not correlated to ROS production at 24 h of exposure. When HTLA-ER cells were treated with **4I** for prolonged periods of time, the higher ROS production, probably due to the failure of the developed antioxidant defences of cells, was well correlated with cell deaths. Based on the findings reported by Brullo et al. [19] and Nambiar et al. [38], we can hypothesize that probably the increased level of ROS caused by **4I** affected the cell-cycle phases by targeting the tubulin system and causing microtubules alteration, thus prompting cells death.

### 2.4. Preparation of P5PA and P5PA-4I Nanoparticles (NPs)

Even if already higher than that of ETO in HTLA-ER NB cells, we enhanced the cytotoxic activity of **4I** by a nanotechnological approach. To this end, **4I**-loaded PA/P5-based NPs (P5PA-4I NPs) were prepared by the slightly modified single emulsification technique described in the literature [39] according to Figure 1.

Analogously, PA/P5-based NPs (P5PA NPs) were obtained without using **4I**. Table 3 collects useful experimental data concerning the preparation of P5PA and P5PA-4I NPs.

Upon sonication at r.t., oily NPs made of 4I, and PA dissolved in dichloromethane (DCM) (yellow spheres in Figure 1) entrapped in P5 (blue circles in Figure 1), were obtained dispersed in the water phase. After removal of DCM by the oil-in-water (O/W) nano-emulsion stabilized by PA, an aqueous dispersion of P5PA-4I NPs was achieved, which was washed with DCM to remove the not encapsulated PA and/or **4I**. Lyophilization of the aqueous dispersion provided solid P5PA-4I NPs. In parallel, we prepared P5PA NPs by the same method and in the same conditions without using **4I**. No additional surfactant or other additive was used, except for the biocompatible fatty acid PA which, by interacting with P5 through hydrogen bonds, stabilized the O/W nano-emulsion [30]. The employed concentration of PA in both cases <30 mM allowed to avoid the emulsion inversion which has reported to occur for PA concentrations >50 mM [30]. P5PA NPs (337.5 mg) and P5PA-4I NPs (402.0 mg) were obtained as pale-brown glassy solids (Appendix A, Appendix A), which were stored in a dryer on P_2_O_5_ for further analyses.

### 2.5. Particles Characterization

We characterized P5PA and P5PA-4I particles by optical microscopy, dynamic light scattering analyses and flow cytometric experiments.

#### 2.5.1. Morphology of P5PA and P5PA-4I by Optical Microscopy

The morphology of P5PA and of P5PA-4I were investigated by optical microscopy (OM) analysis and compared with that of P5 particles. In the performed experiments, first powdery samples were observed (Figure 7), then sonicated aqueous dispersions of P5, P5PA and P5PA-4I were examined (Figure 8).

P5, P5PA, as well as P5PA-4I, when observed in the solid state, appeared as aggregates of large dimensions. However, while P5 appeared to be formed by aggregates of bright crystal-like flakes (Figure 7a,b), spherical protuberances of about 200 µm could be identified in the aggregates of P5PA (Figure 7c,d). On the contrary, P5PA-4I appeared as an array of amorphous aggregates, within which dispersed microparticles of about 70 µm can be identified (Figure 7e–g, evidenced by red arrows). When the sonicated aqueous dispersions of P5, P5PA and P5PA-4I were analysed, separated beads of smaller dimensions, which could be micro-aggregates of the NPs later detected in the DLS analyses (see the following Section 2.5.2), were observed for all samples (Figure 8, evidenced by red circles or red arrows) especially when the meniscus of the water drop on the glass-slide was examined (Figure 8b,c (P5), Figure 8e,f (P5PA NPs), Figure 8h,i,j (P5PA-4I NPs). Interestingly, even if the concentrations of dispersions were similar for all samples, an increasing amount of dispersed material was observed going from the dispersion of P5 (Figure 8a), to that of P5PA (Figure 8d) and then to that of P5PA-4I (Figure 8g). This finding evidenced a higher tendency of P5 to settle down and a greater stability of the aqueous dispersions of P5PA and P5PA-4I due to the stabilizing action of PA [30]. Moreover, while the particles of P5 appeared as spherical beads, those of P5PA and of P5PA-4I were morphologically more irregular. Curiously, during the analyses, after an initial inertia of the dispersed particles, an ever more rapid migration was observed from the centre of the drop towards its meniscus where they accumulated, tending to form larger aggregates. As an example of this phenomenon, a link to a video (Appendix A) has been included here and in Appendix A, showing the described rapid migration of P5PA-4I NPs (https://clipchamp.com/watch/O9s7mm5bxkS), (accessed on 28 September 2023).

#### 2.5.2. Particle Size, Zeta Potential (ζ-p) and Polydispersity Index (PDI) of P5PA and P5PA-4 by DLS Analyses

The hydrodynamic diameter (Z-AVE, nm) and PDI of P5PA and P5PA-4I NPs were determined by DLS analysis, while ζ-p measurements were carried out to determine their surface charge (mV). The results have been reported both in Table 4 and in Figure 9.

While the previously reported particles of P5 alone showed an average size of 334 nm with a PDI of 1.012, those of both P5PA and P5PA-4I showed larger sizes of 516.4 and 541.5 nm and significantly lower DPIs of 0.391 and 0.195, respectively. Probably, the inner oily core made of PA or PA and **4I** decorated in the surface by P5, significantly increased both the usual dimensions of PA NPs obtainable by emulsion techniques (about 100–158 nm) [29,40] and those of P5 NPs previously prepared by radical polymerization and solvent–antisolvent precipitation method [31]. By comparing the size of P5PA-4I NPs with that of empty P5PA NPs, we can assume that the larger size of the first ones was due to the incorporation of **4I** [41]. Anyway, it is known that nano-emulsion techniques could provide nanoparticles in the size range of 20–1000 nm [42] and that PA-based NPs with dimensions in the range 500–567 nm have been reported by Yan et al. [43]. Concerning the significantly lower polydispersity of P5PA and P5PA-4I NPs with respect to P5, it was due to the emulsification method with respect to other methods that allow us to obtain NPs with lower PDI. In this context, PA-modified human serum albumin paclitaxel nanoparticles have been recently reported showing PDI values of about 0.2 [40], while PA-containing NPs were developed by He et al. having PDI in the range 0.120–0.147 [29]. According to this latter study, the presence of PA in NPs caused the reduction of the PDI value of doxorubicin (DOX)-loaded PLGA NPs from 0.259 to 0.147, thus confirming the contribute of PA in reducing the very large PDI of original P5 NPs. Curiously, while the encapsulation of DOX in PLGA/PA NPs reported by He and co-workers caused a significant increasing in the DPI values [29], in our case the encapsulation of **4I** favoured a remarkably decrease in PDI from 0.391 to 0.195. Values of 0.2 and below are most commonly deemed acceptable in practice for polymer-based nanoparticle materials finalized to drug delivery [44]. The ζ-p of both P5PA (+6.97 mV) and P5PA-4I NPs (+8.38 mV) were positive but much lower than the ζ-p of P5 (+58 mV), probably due to the presence of PA, which demonstrated negative values of ζ-p [29,41]. Although polymer particles with high positive ζ-p are reported to be capable of faster absorption on cells’ surface by electrostatic interactions, with a possible easier internalization than particles with negative ζ-p [31], the low ζ-p and cationic character of NPs developed here could reduce the possible activity as membrane disruptors and the cytotoxic effects on normal cells. In fact, it was reported that low positive ζ-p of gelatine nanoparticles loaded with cryptolepine showed low haemolytic effects on human blood [45], while another study found that high positive or negative ζ-p of nanobubbles caused haemolysis of red blood cells [46].

#### 2.5.3. Estimation of Particles Size and Complexity Distributions by Cytofluorimetric Analyses

Cytofluorimetric analyses were performed on clear dispersions of P5, P5PA and P5PA-4I to further investigate their particle size and complexity distributions. Flow cytometry is a potent technology that rapidly analyses single cells or particles suspended in deionized water or in a buffered salt-based solution, while they flow past single or multiple lasers [47]. Each particle can be analysed for both visible light scatter and for one or multiple fluorescence parameters [48]. Concerning visible light scatter, it is measured in two different directions, the forward direction (Forward Scatter or FSC) and at 90° (Side Scatter or SSC). FSC values are proportional to the diameter of the cell or particle, while SSC measurements to their complexity or granularity [48,49]. Light scatter is independent of fluorescence [48]. In our case, a dispersion of unstained polystyrene particles with 1 micron diameter was used as the standard to calculate the diameters of P5, P5PA and P5PA-4I, while Type I deionized water was used as the control. Several types of plots were obtained, which were reported in Appendix A (Appendix A) and in Figure 10, Figure 11 and Figure 12.

Table 5 reports the FSC and SSC statistics (mean and median) calculated and provided by the instrument software for the R2 gated contour plot of SSC-HLog vs. FSC-HLog (panel d in Appendix A and Figure 10, Figure 11 and Figure 12).

The mean and median FSC values of water, P5, P5PA, and P5PA-4I were used to estimate the diameter of their particles against that of standard beads (1000 nm), while the mean and median SSC values were used to estimate the complexity of particles. Interestingly, the diameter of P5 particles perfectly matched that provided by DLS analyses [31]. DLS provided a value slightly higher (334 vs. 317 nm) because it measures the hydrodynamic diameter of particles that are larger than the diameter of the not solvated particles. On the contrary, the size of P5PA and P5PA NPs was significantly higher than that found in the DLS analyses (988 and 1094 vs.516 and 542 nm). This because, while flow cytometry is a static light scattering technique in which single particles scatter light elastically [50], DLS is a dynamic quasi-elastic light scattering technique based on the diffusion properties of particles in a solution over time, which primarily measures their Brownian motion. As a consequence, larger particles tending to settle are not or are barely perceived by the instrument, thus shifting the maximum of the pseudo-gaussian size distribution toward minor dimensions. However, also in the flow cytometry analyses as in the DLS ones, the difference in size between P5PA and P5PA-4I was almost proportionally respected.

Figure 13 shows a comparison between the histogram plots of FSC (proportional to size distributions) (a) and SSC (proportional to materials complexity) (b) of all samples.

Light orange graphs referred to standard beads. Figure 13a evidenced that the size distributions of P5PA and P5PA-4I were very similar to each other with mean and median values like that of standard beads. Anyway, since having a very large coefficient of variation (CV%), their size distributions also comprehended smaller particles, as those detected in the DLS analyses. Analogously, Figure 13b highlighted that the less complex material was P5 consisting only of the monomer M4 and dimethylacrylamide (DMAA) [31], while complexity increases in P5PA, containing M4, DMAA and PA and increases slightly again in P5PA-4I also containing **4I**.

#### Principal Component Analysis (PCA) on Statistics by Flow Cytometry

All statistics data were arranged in a 4 × 29 matrix of 116 variables that was processed by using the principal components analysis (PCA) [51]. PCA allowed us to visualize the reciprocal positions occupied by standard beads, P5, P5PA NPs and P5PA-4I NPs in the scores plot of component 1 (PC1 explaining the 77.9% of variance) vs. component 2 (PC2 explaining the 20.6% of variance) (Appendix A). Samples located close to each other along either component PC1 or PC2 had similar statistics, while those placed far apart had different statistics.

According to the score plot showed in Appendix A, the samples were well-separated both on PC1 and on PC2. On PC1 they were located on the base of their size and complexity, while on PC2 they were clustered on the base of their chemical composition. Particularly, on PC1 going from the bottom of the score plot (negative scores) to the top (positive scores) both the size and the complexity of materials increases. Differently, on PC2, synthetic materials were located at positive scores, while commercial polystyrene beads were positioned at negative scores.

### 2.6. Chemometric Assisted ATR-FTIR Analyses

ATR-FTIR analyses were carried out on **4I**, PA, P5, P5PA NPs and P5PA-4I NPs to have qualitative indication of the chemical structure and composition of P5PA and of P5PA-4I NPs. Figure 14 and Figure 15 show comparisons of the obtained spectra.

In particular, Figure 14 highlighted as in the spectrum of P5PA NPs that we can observe bands that are peculiar for both P5 and PA. From the left side of the spectrum, we found the broad and weak band of the NH stretching of P5 (3600–3200 cm^−1^), two intense bands relative to the CH stretching of the numerous methylene groups of PA (2916 and 2848 cm^−1^), which surmounted those weaker of P5, and two intense bands relative to the C=O stretching of the carboxylic group of PA (1699 cm^−1^) and to the C=O stretching of the amide groups of P5 (1616 cm^−1^). Moreover, in the fingerprint region of the spectrum of P5PA, we can observe many bands belonging to functional groups existing in PA, and not observable in the spectrum of P5, including the bands at 1472, 1430, 940 and 720 cm^−1^. On the contrary, a band non-present in the spectrum of PA but existing in that of P5 is detectable at 1143 cm^−1^, thus further confirming the presence of P5 in the obtained NPs. Figure 15, showing the comparison between the empty NPs, the ingredient (**4I**) and the NPs loaded with **4I**, highlighted that in the spectrum of P5PA-4I NPs (green), in addition to the bands observed in the light blue spectrum of P5PA in Figure 14 (3422, 2916, 2848, 2679, 1699 and 1622 cm^−1^), new bands peculiar of **4I**, such as bands at 3292 (CH aromatic stretching), 1406, 1270, 1140, 1003, 938, 861, 697 and 620 cm^−1^, were observable, thus confirming the success of the encapsulation reaction.

#### Principal Components Analysis (PCA) on the ATR-FTIR Data

As explained in the previous section, just by observing the ATR-FTIR spectra in Figure 14 and Figure 15, the presence of PA with P5 in P5PA NPs and of **4I** in P5PA-4I NPs was unequivocable. Anyway, we found further confirmation processing the ATR-FTIR spectral data of all samples using PCA as previously made with statistics data by cytofluorimetric analyses. PCA allowed us to visualize the reciprocal positions occupied by **4I**, P5, PA, P5PA NPs and P5PA-4I NPs in the scores plot of PC1 (explaining the 64.9% of variance) vs. PC2 (explaining the 21.8% of variance) (Figure 16). In this case, samples located close to each other along either component PC1 or PC2 are structurally similar, while those placed far apart are structurally different.

As observable, PC1 differentiated between empty NPs (P5PA), which were located at positive scores as with their components (P5 and PA) and NPs loaded with **4I**, which were positioned at negative scores as **4I**, thus evidencing structural similarities between the obtained NPs and their ingredients. Anyway, empty and loaded NPs were very close to the zero-score and to each other, evidencing a structural similarity even between them (both are made of P5PA). On PC2, P5PA and P5PA-4I NPs results were placed at score zero, confirming the structural similarity previously evidenced on PC1. The minor distance between both types of NPs and P5 respect to PA, recognized this copolymer as the component loaded in a major amount.

### 2.7. Determination of the Exact Amount of ***4I*** Loaded in P5PA-4I NPs

The exact amount of **4I** contained in an exactly weighted quantity of P5PA-4I NPs was estimated using the **4I** calibration curve constructed as reported in the Experimental Section and as shown in Appendix A.

Once obtained, the quantity of **4I** (mg) present in the total amount of P5PA-4I prepared (mg), it was used to calculate the values of drug loading (DL%) and entrapment efficiency (EE%). To this end, we dispersed P5PA-4I NPs in MeOH, obtaining a fine suspension, which was vigorously stirred to promote the full release of **4I**. The sample was centrifugated and the clear solution was reacted with the Schiff reagent up to a magenta coloration. Considering the amount of **4I** mathematically estimated according to the initial loading (Table 3), aliquots of the obtained coloured solution were diluted 1:10 to remain within the limits of the calibration curve before UV-Vis analysis at 543.88 nm. The obtained Abs values were used to determine the **4I** concentration (mg/mL) by employing the equation shown in Appendix A. Determinations were made in triplicate and reported in Table 6.

The content of **4I** in the total amount of P5PA-4I prepared (402.0 mg) was 83.9 mg, DL (%) was 20.87%, thus confirming the estimated value reported in Table 2 and the EE (%) was 96.55%. In this regard, the EE% of P5PA-4I was very good, comparable to that reported by Sari et al. [52], and close to 100% as reported recently [53]. On the contrary, the DL% was much higher (20.9% vs. 0.74%) than that obtained by the same authors [53], and that reported by other authors for lipid core nano-capsules [54].

### 2.8. Potentiometric Titrations of P5PA and of P5PA-4I NPs

Upon titrations of both P5PA and P5PA-4I NPs dispersed in water as described in the Experimental Section, by plotting the measured pH values vs. the aliquots of HCl 0.1 N added, the titration curves of the samples were obtained as shown in Figure 17a. The first derivative (FD) curves of the titration lines shown in Figure 17b were obtained by computing the values of dpH/dV and plotting them against the corresponding volumes of HCl 0.1 N. The maxima of the FD lines represent the different titration end points. For comparison purposes, P5 alone was titrated in the same conditions as P5PA and P5PA-4I NPs and the titration curve, as well as the related FD are available in Appendix A in the Appendix A file.

The representative titration curve of P5PA (Figure 17a) and related FD (Figure 17b) evidenced that the potentiometric titration of the aqueous solution of P5PA added with NaOH 0.1 N (pH = 10.19) exhibited mainly three successive end points, upon the adding of 1.3, 3.5 and 8.5 mL HCl 0.1N, corresponding to the major maxima of 2.15, 0.86 and 1.66 (Figure 17b). As reported by the group of Zu in 2021, the first end point corresponded to the titration of any excess NaOH [55].

Due to the presence of PA, which at a physiological pH of 7.4 is in the anionic form [56], the second end point (maximum 0.86) occurring at pH = 7.12 corresponded to the titration of palmitate, while the third one can be ascribed to the protonation of the amine groups of P5. In these conditions, the number of millimoles of PA and primary amine groups can be obtained using the volumes of HCl 0.1 N added between the first and the second and between the second and the third end point (2.2 mL and 5.0 mL), respectively. As reported in Table 7 the milliequivalents of PA and those of the primary amine groups were 3.6 ± 0.03 mmol/g and 8.3 ± 0.03 mmol/g, respectively. The milliequivalents of NH_2_ groups contained in 1 g of P5 (12.5 ± 0.03 mmol/g) computed on the basis of its titration results (Appendix A, Appendix A) have been reported in Table 7 for comparison purposes [31]. In this regard, considering that, 1 g of P5PA should contain 0.667 g P5 (Table 2), the milliequivalents of primary amine groups per gram of P5 estimated here were 12.4 mmol/g, perfectly in accordance with the result obtained titrating P5 alone (12.4 vs. 12.5 mmol/g, error = −0.4%).

The titration curve of the aqueous solution of P5PA-4I added with NaOH 0.1 (pH = 9.20) demonstrated an additional maximum, since the groups that can undergo titration are the hydroxyl group deriving by the NaOH added, the carboxylate group of PA, the primary amine groups of P5 and finally the two weakly basic nitrogen atoms (the sp_2_ nitrogen atom of the pyrazole ring and the nitrogen atom of the Schiff base) of **4I**. As shown in Figure 17b, the FD of the titration curve of P5PA-4I revealed four major maxima (0.70, 0.60, 1.60 and 1.58) corresponding to the addition of 1.5, 4.5, 7.5 and 8 mL of HCl 0.1N and representing the titration end points of NaOH, PA, primary amine groups of P5 and the two basic nitrogen atoms of the imidazo-pyrazole. We have used the volume of HCl employed between the first end point and the second one and that employed between the second and the fourth end point to estimate the equivalents of PA and those of the total basic nitrogen atoms of P5 and **4I**, respectively. As reported in Table 7, the milliequivalents of PA and of basic nitrogen atoms were 4.9 ± 0.03 mmol/g and 5.8 ± 0.03 mmol/g, respectively. Additional speculations on possible uses of titration data are available in Appendix A).

#### 2.8.1. Buffer Capacity

Cationic materials such as those developed here and those used to bind and transport genetic material inside the cell for gene-therapy purposes, are reported to be internalized by several methods including endocytosis [57]. Once inside the cells by endocytosis, cationic materials are within the endosome, and need to escape it so as to not be degraded into the lysosomes. It was reported that materials capable of avoiding lysosomal degradation act as a “proton sponge”, attracting protons inside the endosome, thus causing osmotic swelling and bursting [57]. In this context, the proton sponge activity of cationic macromolecules mainly depends on their buffer capacity (β = dV_HCl_)/d(pH)) [58], as well as on their average buffer capacity (β_AVE_) defined as the volume of HCl necessary to cause a variation of pH equal to one unit in the pH range 4.5–7.5 [59]. Particularly, the higher the values of β and β_AVE_, the higher the proton sponge activity, the faster the lysosome escape and the longer the residence time in cells. Thus, to predict the capability of P5PA and P5PA-4I NPs to avoid premature degradation and inactivation, we evaluated their β and β_AVE_ by data of potentiometric titration. The max β values and values of β_AVE_ determined for P5PA and P5PA-4I NPs have been reported in Table 8 and compared with those obtained for P5 and for commercial branched polyethyleneimine (*b*-PEI 25K), a reference standard recognized for having good buffer capacity [60].

By reporting in graphs all the β values calculated for P5, P5PA, P5PA-4I NPs and *b*-PEI vs. the values of corresponding pH, their buffer capacity curves were obtained (Figure 18a and Appendix A), while in Figure 18b the β_AVE_ values computed for all samples have been reported using a bars graph.

As evidenced in Appendix A in Appendix A, *b*-PEI has two maxima of β (Table 8) in the pH range of interest (pH = 6.81 and 7.33), which confer *b*-PEI a buffer capacity suitable to exert an adequate proton sponge activity [60]. Anyway, the maxima of *b*-PEI are not visible in Figure 18a because almost 10-fold lower than the maximum of P5 (0.6667 at pH = 6.10) and extremely lower than the maxima of P5PA and P5PA-4I. Particularly, P5PA demonstrated two maxima at pH = 6.18 and 6.76, while P5PA-4I showed two maxima of equal height (4.5455) and even more intense than those of P5PA at pH = 6.49 and 7.45, thus assuring strong proton sponge activity, rapid lysosomal escape and high residence time in cells. Curiously, as reported in Figure 18b, while the buffer capacity of *b*-PEI was lower than that of P5, its β_AVE_ was more than two-fold higher, thus justifying its reported high efficiency as a proton sponge [60]. However, our P5PA and P5PA-4I NPs demonstrated the highest average buffer capacity as well, thus confirming what was previously evidenced by data of buffer capacity.

#### 2.8.2. Evaluation of the Chemical Stability of P5PA and P5PA-4I NPs Dispersions over Time

To evaluate the chemical stability over time at room temperature of both P5PA and P5PA-4I NPs aqueous dispersions, potentiometric titrations of both materials were carried out after 1, 5, 10 and 15 days from the first one performed, in the same conditions obtaining the related titration data. All obtained titration data (pH values) were arranged in a matrix 24 × 10 of 240 variables, which was subjected to PCA as previously described for cytofluorimetric and ATR-FTIR data. Figure 19 shows the score plot of PC1 (explaining the 97.6% of variance) vs. PC2 (explaining the 1.3% of variance) with the confidence ellipses at 95% (red solid line), 99% (red dashed line) and 99.9% (red dotted line). The confidence ellipses enclosed samples at different levels of confidence, from the more restrictive (95%) to the larger one (99.9%) not significantly different in terms of titration data, thus indicating chemical stability.

Based on the results in Figure 19, on PC1 samples that were clustered in a manner dependent on the presence or not of **4I,** with samples containing **4I** located at negative scores, while those without **4I** were at positive values. Collectively, all samples of both materials were comprised within the more restrictive ellipse, evidencing that no significant difference existed in titrations data measured up to 15 days, thus establishing for good stability of both types of NPs during this monitoring time. Additionally, while P5PA NPs showed samples P5PA_1 and P5PA_5 within the confidence ellipse, but not overlapped with the other samples of the group, no sample in the group of compounds containing **4I** was clearly separated from others, thus indicating very minimal differences and a stability higher than that of P5PA NPs.

To further confirm the absence of outliers, intending compounds with titration data significantly different from those of the first titrated ones, among the samples analysed, we make Q and T^2^ tests using the diagnostic tools of CAT. We obtained both separate Q and T^2^ contribution plots (Appendix A, respectively, in Appendix A), and the influencing plot of T^2^ Hotelling index vs. Q index (Appendix A). All plots evidenced that no outlier existed among the analysed samples, which were located all within the solid lines, indicating the more restrictive critical values (*p* = 0.05). Then, we performed further titrations of our samples after 20 days in aqueous dispersion at room temperature and arranged the measured data in a matrix 2 × 24 of 48 variables that was processed by PCA, obtaining the score plot reported in Appendix A in Appendix A. This new score plot confirmed that P5PA-type samples located at positive scores while the P5PA-4I-type ones at negative scores. We used this new data as an external data set and projected them in the data set of the ten samples previously processed, which acted as a training set representing a population of not-degraded samples. Figure 20 shows the obtained score plot, where the new samples titrated after standing in dispersion for 20 days, have been indicated with the name in red, while the samples of the training set have been represented with empty circles.

Based on the results shown in Figure 20, evidencing both P5PA_20 and P5PA4I_20 located within the most restrictive confidence ellipse (*p* = 0.05), it was established that also after 20 days no significant difference was detected in the titration data, thus further establishing a good stability over time of our materials. Further titrations were performed after 25, 30 and 40 days from the first one and data were treated as just described with similar results.

### 2.9. Evaluation of ***4I*** In Vitro Release Profile

The in vitro release profile of **4I** from P5PA-4I NPs was investigated by the dialysis method as described in the Experimental Section. In parallel, the release of **4I** from a **4I** suspension in PBS was monitored in the same conditions. The obtained concentrations were used to compute the cumulative drug release percentages of **4I** over time, which were plotted vs. times obtaining the curves shown in Figure 21.

As is observable, the release profiles of **4I** from P5PA-4I and from its suspension were very similar and characterized by a burst release within the first four hours. A very slow release was observed up to 24 h, followed by an increment in the release rate, especially from P5PA-4I, in the subsequent four hours leading to a linear sustained release during the subsequent 44 h up to the 72 h of the monitoring. The main differences between the release of **4I** from P5PA-4I NPs and that from the **4I** suspension was the amount of **4I** released. Notably, at any stage the **4I** released from P5PA-4I NPs was about two-fold higher than that released from the suspension, thus indicating that by formulating **4I** in NPs, its aqueous solubility was remarkably improved, thus assuring a higher bioavailability in a possible future oral administration. Collectively, the amount of untreated **4I** released in the acceptor medium at 72 h was only the 45% of the **4I** suspended in the dialysis bag, while that released by NPs was the 93% of the **4I** loaded in NPs, thus assuring a systemic concentration more than two-fold higher than that achievable by not formulated **4I**. Curiously, the release of **4I** from the **4I** suspension in PBS observed this time was very different from that previously observed by us (study non yet published), which was faster and quantitative. Probably, the higher concentration of **4I** in the dialysis bag used here (4.2 mg/mL vs. 0.35 mg/mL) could have led to flocculation and aggregates formation, which hampered the release of **4I** in the acceptor medium. Anyway, such phenomenon did not occur when the same concentration of **4I** was transported by P5PA-4I NPs, thus allowing an almost quantitative release.

#### Kinetic Studies

The kinetics and main mechanisms governing the releases of **4I** were investigated by fitting the data plotted in Figure 21 to the most common kinetic models [61,62].

The obtained dispersion graphs with the related linear regressions provided by Microsoft Excel software 635 using the OLS method and their equations are available in Appendix A in Appendix A. The value of the coefficient of determination (R^2^) associated with each equation was the parameter used to determine which model better fit the release data. The model with the highest R^2^ value was declared the best fitting one. The R^2^ values obtained for all models tested were reported in Table 9, and accordingly, the **4I** release from both P5PA-4I NPs and from the 4I suspension best fit with the Higuchi kinetic model.

The Higuchi kinetics are expressed by Equation (1):(1)$$Qt=Kh\;\times \;t^{n}$$
where *Qt* (variable y of the equation in Appendix A) is the amount of **4I** released at time *t*, *Kh* is the Higuchi kinetic constant, *t* is time, and *n* is the release exponent, which in the Higuchi models is equal to 0.5, thus establishing that the mechanism that governs both the releases of **4I** from NPs and from the **4I** suspension is based on Fickian diffusion. The kinetic constants (*Kh*) corresponded to the slopes of the equations in Appendix A and were equal to 10.12 (P5PA-4I) and 5.27 (**4I** suspension), thus establishing that at the same time point the **4I** released from NPs was about two-times higher than from the suspension, as previously observed in Figure 21.

### 2.10. Cytotoxic Effects of ***4I***, P5PA and of P5PA-4I on NB Cells

Both NB cell populations were exposed to increasing concentrations (0.5–50 μM) of **4I**, P5PA-4I NPs (administered in a dose capable of providing 0.5–50 μM **4I**), and P5PA NPs (administered at the same dose of those contained in the P5PA-4I) for 24, 48 and 72 h and the effect of these treatments on cell viability was evaluated. As shown in Figure 22a,b, when administered at the same concentrations, nanotechnologically manipulated **4I** provided by P5PA-4I NPs was more cytotoxic than pristine **4I**, especially in HTLA-ER cells. Specifically, it reduced the cell viability by a further 10% with respect to pristine **4I**, at any time of exposure tested (Figure 22b). These findings agree perfectly with the release profiles of **4I** from P5PA-4I NPs and from its suspension observed previously (Section 2.9). Specifically, regardless of the time of the experiment, the amount of **4I** released by NPs was always higher than that released by the suspensions. Particularly, at 72 h, the amount of **4I** released by NPs was 93% against the 45% released by the **4I**-suspension, thus assuring a remarkably high concentration of **4I** at the target cells, which was responsible for the higher cytotoxic activity. Additionally, the time dependent cytotoxic activity of P5PA-4I NPs which markedly increased from 42 to 72 h of exposure also depended on the controlled and sustained release profile of **4I** for the nano-formulation. Notably, although P5 [32] and PA [20,21,22,23,24,25] have been reported to have antiproliferative activity, P5PA per se did not induce changes in cell viability (Appendix A). The reason is probably due to the acid-base interaction between the ammonium groups of P5 and the PA, that, at physiological pH of the experimental medium, was in the carboxylate form [56], thus causing a reciprocal inactivation by providing a salt. Therefore, the cationic groups of P5, responsible for its cytotoxic [32] and antibacterial activity [31], were neutralized, determining the activity loss.

The cytotoxic effects of P5PA-4I NPs were further confirmed by calculating the IC_50_ values. In fact, as reported in Table 10, this value was 1.2-, 1.4- and 1.5-fold lower (24, 48 and 72 h, respectively) in HTLA-230 cells treated with P5PA-4I NPs respect to those treated with pristine **4I**. An even greater reduction of IC_50_ values (1.9-, 1.5-, 1.6-fold lower) was observed in HTLA-ER cells treated with the **4I**-enriched NPs compared to their treatment with **4I**.

The application of two sample Student *t* test established that the IC_50_ values of P5PA-4I NPs were significantly lower than those of **4I** in both NB cell populations and at both confidence intervals (*p* = 0.01 and *p* = 0.05), in particular in HTLA-ER cells at all times of exposure. Then, to analyse the efficacy of these compounds in counteracting the chemoresistance of HTLA-ER cells, the cytotoxic activity of **4I** and of P5PA-4I NPs (0–50 µM, Figure 23) were compared with that of ETO reported by Colla [34]. The results are shown in Appendix A and in Figure 23.

As shown in Appendix A, 10 µM **4I** was significantly more cytotoxic than ETO administered at the same dose. Interestingly, a two-sample Student *t* test showed that **4I**-induced effects in HTLA-ER cells, already at 24 h exposure, were significantly higher (*p* = 0.05 and *p* = 0.01) than those induced by ETO. Moreover, the cytotoxic effects of P5PA-4I NPs on HTLA-ER cells were confirmed to be significantly higher than those of **4I** (*p* = 0.05 and *p* = 0.01) and also higher than those of ETO (*p* = 0.05 and *p* = 0.01). In fact, the IC_50_ values of P5PA-4I NPs were 1.9- and 3.3-fold lower than those of **4I** and ETO, respectively. Based on these findings, the embedment of **4I** in P5PA NPs represents a promising nanotechnological strategy to strongly enhance the cytotoxic action of **4I** as well as of other imidazo-pyrazole derivatives [19]. Collectively, empty PAP5 NPs, which have been found deprived of intrinsic cytotoxic effects, could represent a safe novel delivery system capable of increasing the cytotoxic action of entrapped traditional or novel anticancer drugs. In this regard, experiments to verify the level of cytotoxicity of P5PA and P5PA-4I NPs on normal cells are programmed to assess better their clinical applicability.

### 2.11. Effects of ***4I*** and P5PA-4I on ROS Production in NB Cells

As reported in Section 2.2, the effects of pristine **4I** (range 0–50 µM) and P5PA-4I NPs on ROS levels were investigated (Figure 24). P5PA NPs, administered to provide P5PA at the same concentration given by P5PA-4I NPs, induced a decrease in H_2_O_2_ levels (Appendix A).

As shown in Figure 24, P5PA-4I increased H_2_O_2_ levels in a time- and dose-dependent manner, although the effect was more evident at 72 h, because of a controlled and sustained release of **4I** from the nano-formulation. In fact, compared to **4I**-treated cells, H_2_O_2_ levels were increased by a further 30% after 72 h in both cell populations treated with 50 μM P5PA-4I (Figure 24). Notably, the prooxidant effects induced by 24 h ETO exposure (range 0–50 µM) in HTLA-ER cells [34] were compared to those induced by **4I** and P5PA-4I NPs (Figure 24b). These results have been analysed and shown in Figure 25.

Also in this case, when **4I** was administered as a nano-formulation (P5PA-4I NPs), already at 24 h, it caused an increase in H_2_O_2_ levels in HTLA-ER cells not only higher than that caused by the pristine **4I**, but also higher than that caused by ETO, at all concentrations tested, and in particular at low concentrations (0.5-5 µM).

#### Correlation between H_2_O_2_ Production and Cytotoxicity of P5PA-4I NPs

To validate the hypothesis of a possible relationship between the cytotoxic and pro-oxidant effect of P5PA-4I NPs, their correlation was tested by plotting the data of cell viability (%) vs. DCFH positive cells (%) obtained in P5PA-4I NPs-treated HTLA-230 and HTLA-ER cells. The dispersion graphs were reported in Figure 26.

According to results reported in Figure 26, a correlation from discrete to good was shown both in HTLA-230 and HTLA-ER cells. Particularly, a discrete correlation was observed in HTLA-230, regardless of the time of exposure. A better correlation was observed in HTLA-ER cells and mainly for 24 h treatment, thus suggesting that, despite the high antioxidant defence in ETO resistant cells, the cytotoxic effects induced by P5PA-4I NPs were dependent on H_2_O_2_ production. On the contrary, the reduced correlation between peroxide production and cytotoxicity of P5PA-4I NPs at 48 and 72 h exposure suggested an adaptive cell response to H_2_O_2_ and the implication of other factors responsible for cancer cell death. Furthermore, these data were also confirmed by the results obtained with P5PA NPs, which were ineffective in reducing cell viability and enhancing H_2_O_2_ levels (Appendix A).

## 3. Materials and Methods

### 3.1. Chemicals and Instruments

All reagents and solvents were purchased from Merck (formerly Sigma-Aldrich, Darmstadt, Germany) and were purified by standard procedures. 2-2′-azobisisobutirronitrile (AIBN) was crystallized from methanol. Organic solutions were dried over anhydrous magnesium sulphate and were evaporated using a rotatory evaporator operating at a reduced pressure of about 10–20 mmHg. Copolymer P5 was prepared and characterized according to procedures and instruments previously reported [30], while imidazo-pyrazoles **4G** and **4I** were synthesized according to Brullo et al. [19]. Attenuated total reflection-Fourier transform infrared (ATR-FTIR) spectra of the present paper were recorded on a Spectrum Two FT-IR Spectrometer (PerkinElmer, Inc., Waltham, MA, USA). Dynamic Light Scattering (DLS) and Z-potential determinations were performed on the same instrument and with the same modalities previously described [32].

Lyophilization and centrifugation were performed as previously described [32]. Titrations were performed using a HI5522-02 Laboratory Benchtop Instrument pH/mV/ISE and EC/TDS/Salinity/Resistivity Research Grade (HANNA Instruments, Padova, Italy).

### 3.2. Preparation of P5PA and P5PA-4I Nanoparticles (NPs)

PA/polystyrene-based (P5PA) and **4I**-loaded PA/polystyrene-based (P5PA-4I) NPs were synthesized using the single emulsification technique like that described in the literature [39]. Briefly, PA alone (119.1 mg; 0.4645 mmol), or 86.9 mg (0.1859 mmol) of **4I** and PA (115.1 mg; 0.4489 mmol) were dissolved in 12 mL dichloromethane (DCM) to form the oily (O) phase. Secondly, P5 (238.2 mg; 0.047 mmol in the first case and 216.0 mg; 0.042 mmol in the second case) was dispersed in 5 mL ultrapure deionized water, to constitute the water (W) continuous phase, and magnetically stirred for 3 min at room temperature (RT). Next, the aqueous solution of P5 was added to the DCM solution containing **4I** and PA or PA alone, and a VWR^®^ USC-300D, Ultrasonic Cleaning Bath (VWR International S.r.l., Milano, Italy) was used for 30 min to emulsify the system, thus obtaining an oil-in-water (O/W) emulsion. The obtained O/W emulsion was further sonicated at 40 °C for the time necessary to completely evaporate the DCM. The aqueous dispersion of the obtained NPs was washed with DCM 3 times to remove free drugs. Finally, the NPs were freeze-dried for 4 days, and the obtained P5PA NPs (337.5) and P5PA-4I NPs (402.0 mg) were stored in a dryer on P_2_O_5_ at RT until further use.

### 3.3. Particles Characterization

#### 3.3.1. Morphology of P5, P5PA and P5PA-4I NPs

The morphologies of P5, P5PA and of P5PA-4I NPs were here investigated by optical microscopy (OM) analysis. In the performed experiments, powdery samples or aqueous dispersions were observed with a Nikon TMS-F inverted phase contrast microscope equipped with 4 × brightfield objective, 10 × phase contrast objective, LWD 20 × phase contrast objective, LWD 40 × phase contrast objective (Nikon Instruments, Inc., New York, NY, USA). Sequential images were acquired with 4 ×, 10 ×, 20 × and 40 × phase contrast objectives. The video was captured with a 20 × phase contrast objective. The camera for image capture was a Moticam 10+ (10 MP) (MoticEurope S.L.U., Cabrera de Mar, Barcelona, Spain).

The aqueous dispersions of samples were prepared at P5, P5PA and P5PA-4I concentrations of 4.1, 5.3, and 5.1 mg/mL, respectively.

#### 3.3.2. Dynamic Light Scattering (DLS) Analysis

The hydrodynamic size (diameter) (Z-AVE, nm) and polydispersity index (PDI) of P5PA and P5PA-4I particles were determined using Dynamic Light Scattering (DLS) analysis. Z-Ave and PDI measurements were performed in water mQ as medium at max concentration of P5 of 3 mg/mL (pH = 7.4), in batch mode using a low volume quartz cuvette (pathlength, 10 mm). The analysis was performed by a photon correlation spectroscopy (PCS) assembly, equipped with a 50 mW He-Ne laser (532 nm) and thermo-regulated at the physiological temperature of 37 °C. The scattering angle was fixed at 90°. Results were the combination of three 10 min runs for a total accumulation correlation function (ACF) time of 30 min. The hydrodynamic particle size result was volume-weighted and reported as the mean of three measurements ± SD. PDI value was reported as the mean of three measurements ± SD made by the instrument on the sample. The ζ-p was measured at 37 °C in mQ water as a medium, and an applied voltage of 100 V was used. The samples were loaded into pre-rinsed folded capillary cells, and twelve measurements were performed.

#### 3.3.3. Cytofluorimetric Analyses

In the performed experiments the aqueous dispersions of P5, P5PA and P5PA-4I NPs prepared for the previous experiments were diluted 1:8 in type I deionized water 0.22 µM filtered (syringe filters, PVDF, 0.22 µm pour size, S.p.A, Milan, Italy cat#EPSPV2230) and filtered using disposable 40 µM mesh cell strainers (Euroclone S.p.A, Milan, Italy, cat#ET6040). Aliquots of the suspensions were loaded for flow cytometry analysis. Appropriate controls were run in parallel: unstained precision size standard beads 1 µm diameter (Polysciences Europe GmbH, Hirschberg an der Bergstrasse, Germany, cat# 64030-15) and deionized type I water 0.22 µM filtered. Data were acquired on a Guava easyCyte 6 flow cytometer (Merck Millipore, Burlington, MA, USA) and processed using the GuavaSoft 3.1.1 software (Merck Millipore). Table 11 collects the experimental set up and the concentration of particles (Ps) expressed as number of particles per mL.

### 3.4. Chemometric Assisted ATR-FTIR Spectroscopy

FTIR spectra of **4I**, P5PA NPs, P5PA-4I NPs, P5, and PA were recorded directly on the solid samples in attenuated total reflection (ATR) mode using a Spectrum Two FT-IR Spectrometer (PerkinElmer, Inc., Waltham, MA, USA). Acquisitions were made in triplicate from 4000 to 600 cm^−1^, with 1 cm^−1^ spectral resolution, co-adding 32 interferograms, with a measurement accuracy in the frequency data at each measured point of 0.01 cm^−1^, due to the internal laser reference of the instrument. The “find peaks” command of the instrument software was used to obtain the frequency of each band. The matrix of spectral data was subjected to by means of CAT statistical software (Chemometric Agile Tool, free down-loadable online, at: http://www.gruppochemiometria.it/index.php/software/19-download-the-r-based-chemometric-software; accessed on 18 July 2023). In particular, we arranged the FTIR data of the spectra acquired for all the samples in a matrix 3401 × 5 (*n* = 17,005) of measurable variables. For each sample, the variables consisted of the values of absorbance (%) associated with the wavenumbers (3401) in the range 4000–600 cm^−1^.

### 3.5. Content of ***4I*** in P5PA-4I, Drug Loading (DL%) and Entrapment Efficiency (EE%)

To construct the **4I** calibration curve a stock solution of **4I** (0.5 mg/mL) was prepared dissolving **4I** in methanol (MeOH). The obtained colourless solution was added with 500 µL of Schiff′ fuchsin-sulphite reagent (Merck Italy, Milan, Italy) and left at room temperature up to reach a strong magenta coloration. Upon proper dilutions with MeOH, standard solutions at **4I** concentrations of 0.025, 0.0375, 0.050, 0.075, 0.100 and 0.125 mg/mL were prepared. The obtained **4I** solutions were analysed using a Fiber Optic UV-Vis Spectrometer System Ocean Optics USB 2000 (Ocean Optics, Inc., Dunedin, FL, USA) in 3 mL quartz cuvettes, thus detecting the related absorbance (Abs) at room temperature and at ʎ abs = 543.88 nm, using a sample of pure MeOH treated with the Schiff reagent, as blank. Determinations were made in triplicate and results were reported as mean of three independent experiments ± SD. The **4I** concentrations were plotted vs. the Abs values, and the **4I** calibration curve (Figure 17) was obtained by least-squares linear regression analysis using Microsoft Excel software. Equation (2) of the developed linear calibration model was the following.
(2)y=9.3246x+0.0531

In Equation (2), *y* was the Abs values measured at ʎ abs = 543.88 nm and *x* the **4I** standard concentrations analysed. In Equation (2), the slope represents the coefficient of extinction (ε) of the Schiff base adduct between **4I** and the Schiff reagent.

To estimate the **4I** contained in P5PA-4I NPs, 50.0 mg of P5PA-4I NPs were dispersed in 10 mL of MeOH and vigorously stirred for ten minutes to promote the release of **4I**. After centrifugation, the clear solution was added with 500 µL of Schiff′ fuchsin-sulphite reagent (Merck Italy, Milan, Italy) and left at room temperature up to reach a magenta coloration. The coloured solution was diluted 1:10 and the content of **4I** was quantified at ʎ max = 543.88 nm by UV-Vis analysis, using a Fiber Optic UV-Vis Spectrometer System Ocean Optics USB 2000 (Ocean Optics, Inc., Dunedin, FL, USA) in 3 mL quartz cuvettes. The solutions were analysed against a methanol solution not containing **4I** and treated with the Schiff reagent as blank. Determinations were made in triplicate and results were reported as mean of three independent experiments ± standard deviation (SD).

The values of *DL%* and *EE%* also indicted as drug loading capacity (*DLC%*) of P5PA-4I NPs were calculated from the following Equations (3) and (4) [63,64].
(3)$$DL\;\left(\%\right)=\frac{Weight\;of\;4I\;in\;NPs}{Weight\;of\;NPs}\times 100$$
(4)$$EE\;\;\left(\%\right)=\frac{Weight\;of\;4I\;in\;NPs}{Weight\;of\;4I}\times 100$$

### 3.6. Potentiometric Titration of R4HG and R4HG-4I

Potentiometric titrations were performed on P5PA and P5PA-4I NPs at room temperature. Samples of each compound were exactly weighed (60.6 mg of P5PA and 60.8 mg of P5PA-4I) and suspended Milli-Q water (m-Q), qb to cover the pH-meter electrode. The solutions showed pH = 7.45 (P5PA) and pH = 7.50 (P5PA-4I). Then, they were treated under magnetic stirring with a standard 0.1 N NaOH aqueous solution (3.0 mL, pH = 10.19 for P5PA and 2.0 mL to pH = 9.20 for R4HG-4I), and potentiometrically titrated under stirring by adding aliquots of 0.2–0.7 mL of HCl 0.1 N for a total volume of 10.0 mL. Additional titrations were carried out after 1, 5, 10, 15, 20, 30, 35 and 40 days to assess the chemical stability of our materials in aqueous dispersion upon time at room temperature.

Titrations were performed in triplicate and measurements were reported as mean ± SD. The titration curves shown in the Discussion Section are those obtained by plotting the data obtained by carrying out the representative experiment described here.

### 3.7. Evaluation of ***4I*** In Vitro Release Behaviors

The in vitro release of **4I** from P5PA-4I and of **4I** from a **4I** suspension in PBS was determined by the dialysis method. About 200 mg of nanocomposite (P5PA-4I), whose content of **4I** determined by the value of DL% (20.87%) was about 42 mg, and 42 mg of **4I** were weighted. The samples were immersed separately in 10 mL of PBS medium (pH = 7.4) in dialysis bags with 3.5 K MWCO (Cellu Sep H1, Orange Scientific, Braine-l’Alleud, Belgium) and subsequently dialyzed against 50 mL of release medium (PBS) at 37 °C and 100 rpm. At fixed interval points (0, 1, 2, 4, 24, 28, 48 and 72 h), aliquots of 1 mL of the release medium were taken out and 1 mL fresh PBS were replenished. Each aliquot was diluted 1:10 with MeOH, added with 500 µL of Schiff reagents and left to react up to a magenta coloration appeared. The amount of **4I** in the aliquots was detected by measuring their absorbance at 543.88 nm using the UV–VIS spectrophotometer previously described and the **4I** standard calibration curve previously constructed. A PBS solution not containing **4I** and treated with the Schiff reagent was used as blank. Determinations were made in triplicate and results were expressed as mean of three independent experiments ± SD. The obtained concentrations were used to compute the cumulative drug release percentage, according to Equation (5).
(5)$$CDR\;\left(\%\right)=\frac{Dt}{Di}\times 100$$

In Equation (5), *CDR* (%) is the cumulative drug release, *Dt* was the cumulative **4I** concentration at time *t*, and *Di* was the initial concentration of **4I** in the dialysis bags (4.2 mg/mL for both P5PA-4I and for the **4I** suspension).

### 3.8. Biological Experiments

#### 3.8.1. Cell Culture Conditions

HTLA-230 human stage-IV NB cells were kindly provided by Dr. L. Raffaghello (G. Gaslini Institute, Genoa, Italy). The HTLA-ER cells were selected by treating HTLA-230 parental cells for six months with increasing concentration of etoposide, as previously reported [34]. Both cell populations were maintained in RPMI 1640 medium (Euroclone Spa, Pavia, Italy) supplemented with 10% Fetal Bovine Serum (FBS, Euroclone Spa, Pavia, Italy), 1% L-Glutamine (Euroclone Spa, Pavia, Italy) and 1% Penicil-lin/Streptomicin (Euroclone Spa, Pavia, Italy) and grown in standard conditions (37 °C humidified incubator with 5% CO_2_).

#### 3.8.2. Cell Treatments

To determine the biological actions of **4I** or **4G**, time- and dose-dependent experiments were carried out, by treating both cell populations for 24, 48 and 72 h, with increasing concentrations (0.5–100 μM) of the compounds. In another series of the experiments, NB cells were treated for 24, 48 and 72 h with increasing concentration (0.5–50 μM) of **4I** or with the appropriate amounts of P5PA or P5PA-4I to provide cells the same concentrations of **4I** and P5PA as described in Table 12. The stock solutions of these compounds were prepared in 40,000-fold diluted DMSO, and pilot experiments demonstrated that the final DMSO concentrations did not change any of the cell responses analysed.

#### 3.8.3. Cell Viability

Cell viability was determined by using the CellTiter 96^®^ AQueous One Solution Cell Proliferation Assay (Promega, Madison, WI, USA) as previously described [35,36].

#### 3.8.4. Evaluation of Hydrogen Peroxide (H_2_O_2_) Production

The production of H_2_O_2_ was evaluated using 2′-7′-dichlorofluorescein-diacetate (DCFH-DA; Merk Life Science S. r. l., Milan, Italy) as previously reported [35,36].

### 3.9. Statistical Analyses

All biological data are expressed as means ± SEM of at least three independent experiments in which six different wells were analysed every time for each experimental condition. Concerning the experiments reported in Section 2.1 and Section 2.2, statistical significance of differences was determined by one-way analysis of variances (ANOVA); *p* < 0.05 was considered statistically significant. Concerning experiments discussed in Section 2.10 and Section 2.11, differential findings among the experimental groups were determined by a two-way ANOVA analysis of variance, with Bonferroni post tests, using GraphPad Prism 5 (GraphPad Software v5.0, San Diego, CA, USA). Asterisks or other indicators (see figures captions) indicate the following *p*-value ranges: * = *p* < 0.05, ** = *p* < 0.01, *** = *p* < 0.001.

The statistical significance of the slope of the **4I**-calibration curve was investigated through the analysis of variance (ANOVA), performing the Fischer test. Statistical significance was established at the *p*-value < 0.05.

## 4. Conclusions

The aim of this study has been to identify a new potential approach for treating chemo-resistant NB and to this end we have industrialized an imidazo-pyrazole enriched drug delivery system by a nanotechnological approach, and we have tested its activity on ETO-sensitive (HTLA-230) and ETO-resistant (HTLA-ER) NB cells. As described in the manuscript, we have used PA as a nano-emulsion stabilizer and P5 as an encapsulating agent and solubilizer, thus achieving P5PA-4I solid NPs. No additives or additional surfactants, except for PA were used to prepare NPs, thus avoiding undesirable and detrimental interactions of other chemicals with **4I** or P5, as well as unpleasant side effects in a possible future use in vivo. The characterization of NPs has shown that they have low positive Z-potentials (+6.97 and +8.53 mV) able to assure low haemolytic toxicity. In addition, the chemometric-assisted ATR-FTIR analyses, the release experiments, the NH_3_^+^ groups determinations and the buffer capacity calculations fully support the physicochemical suitability of P5PA-4I NPs as an efficient new drug delivery system. Although further experiments are needed to optimize the nano-formulation here reported, including cytotoxicity investigations on normal cells, the embedment of **4I** in P5PA NPs can represent a promising nanotechnological strategy to strongly enhance the cytotoxic action of **4I** as well as of other imidazo-pyrazole derivatives. Collectively, the empty PAP5 NPs, deprived of intrinsic cytotoxic effects, could represent a safer and biocompatible platform for the design of novel delivery systems capable of increasing the cytotoxic action of entrapped traditional or novel anticancer drugs.

## Data Availability

All data supporting the reported results are included in the present manuscript and in the associated Appendix A.

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
