# Peer review of "Imidazo-Pyrazole-Loaded Palmitic Acid and Polystyrene-Based Nanoparticles: Synthesis, Characterization and Antiproliferative Activity on Chemo-Resistant Human Neuroblastoma Cells"

_ijms, 2023, doi:10.3390/ijms241915027_

Round 1
Reviewer 1 Report
The authors developed 4I-loaded palmitic acid and polystyrene-based cationic nanoparticles with high drug loading and encapsulation efficiency by a single oil-in-water emulsification technique. They also performed thorough characterization of the particles to demonstrate its efficacy. However, following points can be considered for further improvements.
1. It would be good if you figure 7 and figure 8 can be improved with appropriate indication of distinctive feature.
2. It would be better if few lines can be added to correlate the release profile of API from NP with the In vitro activity when tested in cells.
3. The score plot used to describe the chemical stability in 2.7.2 section may be confusing to general readers. A more general interpretation, e.g. X% amount API was stable even after Y amount of time when stored in Z degree temperature.
Author Response
The authors developed 4I-loaded palmitic acid and polystyrene-based cationic nanoparticles with high drug loading and encapsulation efficiency by a single oil-in-water emulsification technique. They also performed thorough characterization of the particles to demonstrate its efficacy. However, following points can be considered for further improvements.
- It would be good if you figure 7 and figure 8 can be improved with appropriate indication of distinctive feature.
We thank the Reviewer for his/her suggestion. As asked, appropriate indications of distinctive features have been inserted in Figure 7 (red arrows) and in Figure 8 (red arrows and red circles) evidencing the particles or the particles aggregates.
- It would be better if few lines can be added to correlate the release profile of API from NP with the In vitro activity when tested in cells.
We thank a lot the Reviewer for his/her comment, which enabled us to improve the discussion on cytotoxic activity of P5PA-4I NPs. As suggested, the correlation between the release profile of 4I form NPs and its biological effects has been evidenced and discussed (lines 738-746 and 808-809).
- The score plot used to describe the chemical stability in 2.7.2 section may be confusing to general readers. A more general interpretation, e.g. X% amount API was stable even after Y amount of time when stored in Z degree temperature.
We thank the Reviewer for his/her comment, but nowadays PCA is a well known chemometric tool applied to a wide variety of data sets provided by different experiments (FTIR, UV-Vis, NIR, potentiometric titrations, NMR, fluorimetry etc.) in several sectors to have reliable information, including quality control, stability control, structural information, conformity et. We have retained to use PCA-assisted potentiometric titrations for assessing the stability of our samples because original and appealing. Also, the method suggested by the Reviewer would not have been suitable for evaluating the stability of empty nanoparticles (P5PA) that do not contain 4I.
Reviewer 2 Report
In the current study the authors synthesized, characterized and determined the citotoxicity on chemo-resistant human neuroblastoma cells of some imidazo-pyrazole-loaded palmitic acid and polystyrene-based nanoparticles. The obtained results are promising, so the 4I-based nano-formulation could represent a new promising macromolecular platform to develop a new delivery system able to increase the cytotoxicity of the anticancer drugs with less side effects. Some suggestions:1. Introduction: It would be logical to present the incidence of neuroblastoma not only for the USA, but also for the whole world or for example for Europe (where the authors are from).
2. Why did you choose in you study etoposide and not another standard chemoteherapeutic (Table 1). You wrote that etoposide is administrated in combination with carboplatin and melphalan (**Table 1).
3. In my opinion the informations presented in lines 87-125 must be presented at methods – point 3.1. (the aspects concerning the synthesis and Figs 1 and 2) and at discussions (from line 107 to 125).
At point 3.1 you must describe the synthesis of 4G, 4I and how you prepared the imidazo-pyrazole-loaded palmitic acid and polystyrene-based nanoparticle.
4. The informations concerning the statistical analysis must be written together (now they are at pg 32 and 35). 5. pg 5, please add: - Table 2: the IC50 values for ETO. - Some other explanations regarding the better activity of compound 4I compared to 4G (besides the IC50 values)
6.pg 32, line 814.: Please add how much DMSO. Did you used another solvent to perform the biologic activity?
7. pg 8: What type of microscope did you use? Please add.8. Please improve the quality of the figures 14 and 15. 9. pg 27, line 698: Why are you referring to the antibacterial activity?
10.You specified that ”further experiments are needed to optimize the nano-formulation here reported”. In my opinion it would be interesting to add how you intend to continue your studies.
Minor editing of English language is required
Author Response
In the current study the authors synthesized, characterized and determined the citotoxicity on chemo-resistant human neuroblastoma cells of some imidazo-pyrazole-loaded palmitic acid and polystyrene-based nanoparticles. The obtained results are promising, so the 4I-based nano-formulation could represent a new promising macromolecular platform to develop a new delivery system able to increase the cytotoxicity of the anticancer drugs with less side effects. Some suggestions:
- Introduction: It would be logical to present the incidence of neuroblastoma not only for the USA, but also for the whole world or for example for Europe (where the authors are from).
We thank the Reviewer for his/her suggestion. In this regard, we have presented the incidence of neuroblastoma in the whole world and in Italy, which is our country (lines 49-51).
- Why did you choose in you study etoposide and not another standard chemoteherapeutic (Table 1). You wrote that etoposide is administrated in combination with carboplatin and melphalan (**Table 1).
We used etoposide, because, as reported in the manuscript, the resistant cells used in the study to test 4I and P5PA-4I NPs are etoposide-resistant and were selected by us upon prolonged treatment with etoposide [Ref. 36].
- In my opinion the informations presented in lines 87-125 must be presented at methods – point 3.1. (the aspects concerning the synthesis and Figs 1 and 2) and at discussions (from line 107 to 125).
We thank the Reviewer for his/her suggestion, but, as also suggested by IJMS, the readers should be able to immediately know the structures of the compounds mentioned, without having to scroll forward in the manuscript to find out their structure. We therefore ask the Reviewer not to let them move. On the contrary, as suggested, lines 106-130 (revised version) have been moved in the Discussion section (lines 132-156)
At point 3.1 you must describe the synthesis of 4G, 4I and how you prepared the imidazo-pyrazole-loaded palmitic acid and polystyrene-based nanoparticle.
We thank the Reviewer for his/her indication. Concerning the synthesis of 4G and 4I, it was already reported in our previous work (Ref.19 cited in the text), so that it would be redundant reporting it in this paper again. Anyway, according to the Reviewer suggestion, the preparation and characterization of nanoparticles (lines 1037-1175 in the revised manuscript) have been moved before the biological part. Please see lines 857-990 (revised version).
- The informations concerning the statistical analysis must be written together (now they are at pg 32 and 35).
As suggested by the Reviewer, the information concerning the statistical analysis has been written in a single Section 3.9. (lines 1023-1035).
- pg 5, please add:
- Table 2: the IC50 values for ETO. - Some other explanations regarding the better activity of compound 4I compared to 4G (besides the IC50 values).
As asked the IC50 of ETO has been added in the manuscript, but in Table 10, i.e. in the section where a direct comparison between 4I, P5PA-4I NPs and ETO is discussed. Additional explanations regarding the better activity of compound 4I compared to 4G have been included in lines 182-190.
6.pg 32, line 814.: Please add how much DMSO. Did you used another solvent to perform the biologic activity?
As asked, the indication of how much DMSO was used has been added in the text. Please see line 1009. No other solvent has been used in biological experiments.
- pg 8: What type of microscope did you use? Please add.
The required information was already present in the original text in the old Section 3.4.1. and now Section 3.3.1. in the revised manuscript (lines 874-882).
- Please improve the quality of the figures 14 and 15.
The quality of Figure 14 and Figure 15 has been improved.
- pg 27, line 698: Why are you referring to the antibacterial activity?
Because P5 has been previously reported for having both cytotoxic and antibacterial effects depending mainly on its cationic character.
10.You specified that ”further experiments are needed to optimize the nano-formulation here reported”. In my opinion it would be interesting to add how you intend to continue your studies.
The information requested has been included (lines 793-795).
Comments on the Quality of English Language
Minor editing of English language is required
All manuscript has been checked to reduce all typos and grammatical errors. Additionally, it was revised by our colleague Prof. Deirdre Kantz, English teacher mother tongue working for the University of Genoa and Pavia, where she teaches scientific English in the degree courses in Pharmacy and Pharmaceutical Chemistry and Technology.